# A nationwide questionnaire study of post-acute symptoms and health problems after SARS-CoV-2 infection in Denmark

Anna Irene Vedel Sørensen [1 ✉], Lampros Spiliopoulos [2], Peter Bager[1,2], Nete Munk Nielsen[2,3], Jørgen Vinsløv Hansen[2], Anders Koch [1,4,5], Inger Kristine Meder [2], Steen Ethelberg [1,5] & Anders Hviid [2,6]

A considerable number of individuals infected with SARS-CoV-2 continue to experience symptoms after the acute phase. Here, we report findings from a nationwide questionnaire study in Denmark including 61,002 RT-PCR confirmed SARS-CoV-2 cases and 91,878 test-negative controls aged 15-years or older. Six to twelve months after the test, the risks of 18 out of 21 symptoms were elevated among test-positives. The largest adjusted risk differences (RD) were observed for dysosmia (RD = 10.92%, 95% CI 10.68–11.21%), dysgeusia (RD = 8.68%, 95% CI 8.43–8.93%), fatigue/exhaustion (RD = 8.43%, 95%CI 8.14–8.74%), dyspnea (RD = 4.87%, 95% CI 4.65–5.09%) and reduced strength in arms/legs (RD = 4.68%, 95% CI 4.45–4.89%). During the period from the test and until completion of the questionnaire, new diagnoses of anxiety (RD = 1.15%, 95% CI 0.95–1.34%) or depression (RD = 1.00%, 95% CI 0.81–1.19%) were also more common among test-positives. Even in a population where the majority of test-positives were not hospitalized, a considerable proportion experiences symptoms up to 12 months after infection. Being female or middle-aged increases risks.

[1] Infectious Disease Epidemiology and Prevention, Statens Serum Institut, 2300 Copenhagen S, Denmark. [2] Department of Epidemiology Research, Statens Serum Institut, 2300 Copenhagen S, Denmark. [3] Focused Research Unit in Neurology, Department of Neurology, Hospital of Southern Jutland, University of Southern Denmark, 6200 Aabenraa, Denmark. [4] Department of Infectious Diseases, Rigshospitalet University Hospital, 2100 Copenhagen Ø, Denmark. [5] Department of Public Health, Global Health Section, University of Copenhagen, 1014 Copenhagen K, Denmark. [6] Pharmacovigilance Research Centre, Department of Drug Design and Pharmacology, University of Copenhagen, 2100 Copenhagen Ø, Denmark. ✉email: aivs@ssi.dk

A significant number of individuals infected with SARS-CoV-2 continue to experience symptoms after the acute phase of infection[1]. These symptoms have collectively been known under many different names including long-COVID, and have now been included in the WHO International Classification of Diseases under the name post-COVID-19 condition[2]. Recently, the WHO established the following clinical case definition: "Post COVID-19 condition occurs in individuals with a history of probable or confirmed SARS-CoV-2 infection, usually 3 months from the onset of COVID-19 with symptoms that last for at least 2 months and cannot be explained by an alternative diagnosis"[3]. The global prevalence of this condition has been estimated to 0.43 (0.34 in non-hospitalized individuals), but results obtained in individual studies vary considerably (0.09−0.81)[4].

The symptomatology of post-COVID-19 condition is complex with the possible involvement of multiple organ systems. A growing number of studies support that in addition to a wide range of unspecific physical symptoms, post-acute COVID symptoms may also comprise impaired cognition, mental health problems, and chronic fatigue-like conditions[5–7]. However, knowledge gaps remain regarding the prevalence, range, and duration of these symptoms in the general population of infected and if subgroups particularly prone to post-acute symptoms exist.

A nationwide study was conducted to provide needed insights into post-COVID-19 conditions. This is to the best of our knowledge, the largest questionnaire survey to date on long-COVID both globally and in the Danish population, the EFTER-COVID (Danish for AFTER-COVID) survey. In this report, we present results based on completed questionnaires from participants with a positive test for SARS-CoV-2 in the period September 2020 to April 2021 and corresponding test-negative controls.

The main objectives of the present study were to: (1) Estimate the risk differences between SARS-CoV-2 test-positive and test-negative individuals for acute as well as post-acute symptoms 6–12 months after the test, (2) Evaluate the duration of post-acute symptoms, and (3) Explore the influence of age, sex, and disease severity (hospitalization) on post-acute symptoms.

## Results
**Participants.** In this study, 430,173 individuals (40.0% test positive) were invited to complete the questionnaire. A total of 153,412 (35.7%) participants fully completed the questionnaire, 16,125 (3.7%) partially completed the questionnaire, whereas 260,637 (60.7%) individuals were non-responders. Among those, who completed the questionnaire, 532 test negatives were excluded, due to reporting having been found seropositive, leaving replies from 152,880 participants available for analysis. The questionnaires were completed approximately 6 (14.7%), 9 (69.7%), and 12 months (15.5%) after the test.

Compared to non-responders, participants who fully completed the baseline questionnaire were more often: females, born in Denmark, older (50–70 years old), more often working within healthcare, and living outside of the capital region (Supplementary Table 1).

Among the 171,992 test positives and 258,181 test negatives, who were invited to participate, response rates were very similar, 35.5% and 35.8%, respectively. The participants consisted of 93,494 females (61.2%) and 59,386 males (38.8%) with median ages 50 years (IQRs: 36, 60) and 54 years (IQRs: 41, 64), respectively (Table 1). Compared to the test negatives, test positives were more often: males, younger, students or having full-time employment, and more physically active, and less often: pensioners or smokers (Table 1).

At least one comorbidity was reported by 36.6% of participants (Supplementary Table 1).

**Symptoms around the test date (acute symptoms).** Among test positives, 84.3% reported at least one acute symptom within the period lasting from 1 week before the test and until 4 weeks after the test with a median of six symptoms, compared to a median of four among test negatives with symptoms as test indication. Among all test negatives, irrespective of test indication, 13.5% reported at least one symptom around the test date with a median of two different symptoms. The most common acute symptoms among test positives were fever (55.0%), fatigue/exhaustion (47.2%) and headache (44.1%) (Supplementary Fig. 1). The largest risk differences (RD) between test positives and -negatives tested due to symptoms, were observed for dysgeusia (altered/reduced sense of taste) (RD = 34.49%, 95% CI 33.74–35.28%), dysosmia (altered/reduced sense of smell) (RD = 33.87%, CI 95% 33.06–34.73%) and fever (RD = 23.90%, 95% CI 22.35–25.28%) (Supplementary Fig. 1).

**Symptoms 6–12 months after test (post-acute symptoms).** Among test positives, 29.6% reported at least one symptom 6–12 months after testing compared to 13.0% of all test negatives. In both groups, two were the median number of symptoms reported. The three most common symptoms 6–12 months after testing positive were fatigue/exhaustion (11.1%), dysosmia (10.9%), and dysgeusia (8.8%) (Fig. 1). The most marked risk differences between test positives and test negatives 6–12 months after test were for dysosmia (RD = 10.92%, 95% CI 10.68–11.21%), dysgeusia (RD = 8.68%, 95% CI 8.43–8.93%), and fatigue/exhaustion (RD = 8.43%, 95% CI 8.14–8.74%) (Fig. 1). In addition, dyspnea, reduced strength in legs/arms, sleeping legs/arms, muscle/joint pain, headache, dizziness, chest pain, reduced appetite, hot flushes/sweat, chills, fever, nausea, diarrhea, abdominal pain, and red runny eyes were all significantly more common among test positives (Fig. 1).

**New diagnoses and general health problems 6–12 months post test.** At least one diagnosis of depression, anxiety, chronic fatigue symptom (CFS), fibromyalgia, or post-traumatic stress disorder (PTSD) with new onset within the first 6, 9, or 12 months after the test was reported by 7.2% of test positives, compared to 3.3% of test negatives. The most frequently reported diagnoses were chronic fatigue syndrome (4.0%), depression (3.5%), and anxiety (3.4%) (Fig. 2). All three diagnoses were more common among test positives compared to test negative with statistically significant risk differences of 2.53% (2.35–2.71%), 1.00% (95% CI 0.81–1.19%), and 1.15% (95% CI 0.95–1.34%), respectively (Fig. 2). PTSD was also marginally more common among test positives with a statistically significant risk difference of 0.16% (95% CI 0.03–0.28%).

Among test positives, 53.1% reported at least one of the following problems with new onset within the first 6, 9, or 12 months after the test date: difficulties concentrating; memory issues; mental exhaustion; physical exhaustion or sleep problems, whereas the proportion among test negatives was 11.5%. The most common problems among test positives were physical exhaustion (RD = 40.45%, CI 95% 39.99–40.97%), mental exhaustion (RD = 32.58%, 32.11–33.09%), difficulties concentrating (RD = 28.34%, CI 95% 27.91–28.78%) and memory issues (RD = 27.25%, CI 95% 26.80–27.71%) (Fig. 3). All the aforementioned health problems were more often reported by test positives than test negatives with large risk differences (Fig. 3).

**Duration of individual symptoms.** When looking at estimated RDs for questionnaires completed at 6, 9, or 12 months separately, RDs tended to decrease over time. Among the ten symptoms with the largest overall RDs, the estimates decreased over

**Table 1 Characteristics of 152,880 participants tested for SARS-CoV-2, September 1, 2020–April 2, 2021.**

| | Positive (*n* = 61,002) | Negative (*n* = 91,878) | *P* value |
|---|---|---|---|
| Age (years) | | | |
| Median (IQR) | 49 (34, 60) | 53 (40, 62) | <0.001 |
| BMI (kg/m²) | | | |
| Median (IQR) | 25.2 (22.7, 28.5) | 25.3 (22.7, 28.6) | 0.45 |
| Sex (*n*, %) | | | |
| Female | 35,830 (58.7%) | 57,664 (62.8%) | <0.0001 |
| Male | 25,172 (41.3%) | 34,214 (37.2%) | |
| Education (*n*, %) | | | |
| Higher (2–4 years, BSc) | 19,078 (31.3%) | 30,105 (32.8%) | <0.0001 |
| Higher (>5 years, MSc, PhD) | 10,439 (17.1%) | 14,692 (16.0%) | |
| Vocational training | 10,223 (16.8%) | 16,785 (18.3%) | |
| General secondary or vocational secondary | 6996 (11.5%) | 7985 (8.7%) | |
| Higher (1–2 years, vocational academy) | 6439 (10.6%) | 10,489 (11.4%) | |
| Primary or elementary school (9th–10th grade) | 5734 (9.4%) | 8734 (9.5%) | |
| Do not know/do not wish to answer | 2092 (3.4%) | 3087 (3.4%) | |
| Employment (*n*, %) | | | |
| Employed full-time | 33,516 (54.9%) | 47,717 (51.9%) | <0.0001 |
| Pensioner or early retiree | 8874 (14.5%) | 17,281 (18.8%) | |
| Employed part-time | 5457 (8.9%) | 9956 (10.8%) | |
| Student | 5833 (9.6%) | 6596 (7.2%) | |
| Self-employed | 3494 (5.7%) | 4207 (4.6%) | |
| Other | 1770 (2.9%) | 3194 (3.5%) | |
| Unemployed or seeking job | 939 (1.5%) | 1205 (1.3%) | |
| Long-term sick leave | 446 (0.7%) | 791 (0.9%) | |
| Stay-at-home parent or on parental leave | 465 (0.8%) | 685 (0.7%) | |
| Benefits recipient | 207 (0.3%) | 246 (0.3%) | |
| Smoking (*n*, %) | | | |
| Never | 31,443 (51.5%) | 44,198 (48.1%) | <0.0001 |
| Not in the past 5 years | 15,739 (25.8%) | 25,225 (27.5%) | |
| Occasionally | 5179 (8.5%) | 6382 (6.9%) | |
| Daily (more than ten cigarettes/day) | 1915 (3.1%) | 5114 (5.6%) | |
| Yes, within the past 5 years | 3390 (5.6%) | 4615 (5.0%) | |
| Daily (less than ten cigarettes/day) | 2357 (3.9%) | 4832 (5.3%) | |
| E-cigarettes/vaping | 806 (1.3%) | 1204 (1.3%) | |
| No information | 173 (0.3%) | 308 (0.3%) | |
| Physical activity—past 6 months (*n*, %) | | | |
| Walk, cycle or light exercise (at least four times/week) | 35,920 (58.9%) | 58,848 (64.1%) | <0.0001 |
| Work out or do gardening (at least four times/week) | 15,163 (24.9%) | 18,490 (20.1%) | |
| Read, watch TV or other sedentary lifestyle | 6742 (11.1%) | 11,696 (12.7%) | |
| Hard training or competitive sports (several times/week) | 3173 (5.2%) | 2840 (3.1%) | |
| Physical form—past 6 months (*n*, %) | | | |
| Good | 25,003 (41.0%) | 33,536 (36.5%) | <0.0001 |
| Fair | 21,999 (36.1%) | 37,288 (40.6%) | |
| Less good | 7010 (11.5%) | 13,109 (14.3%) | |
| Really good | 5230 (8.6%) | 4602 (5.0%) | |
| Poor | 1760 (2.9%) | 3343 (3.6%) | |

Notes: *P* values were estimated using student's *t* test for continuous variables and Pearson's Chi-squared test for categorical variables. No adjustments for multiple comparisons were made. Detailed *P* values: age: *P* = 6.39E-321; BMI: *P* = 0.448; sex: *P* = 2.68E-56; Education: *P* = 1.75E-86; employment: *P* = 3.43E-209; smoking: *P* = 3.04E-187; physical activities: *P* = 2.59E-223; physical form: *P* = 3.67E-299.
An individual could only participate in the study once, as either test-positive or test negative.

time for all except dysosmia and dysgeusia for which estimates were largest at 9 months (Supplementary Table 3).

**Post-acute symptoms among hospitalized patients**. The occurrence of post-acute symptoms among test positives hospitalized due to covid-19 (4.0%) and non-hospitalized test-positive individuals (96.0%) was compared (Supplementary Fig. 2). Considerable risk differences were observed for fatigue/exhaustion (RD = 8.64%, 95% CI 6.70–10.74%), reduced strength in arms/legs (RD = 7.13%, 95% CI 5.55–8.66%) and dyspnea (RD = 6.71%, 95% CI 5.17–8.39). The risk for all symptoms, except for dysgeusia, dysosmia, and runny nose were higher among hospitalized than non-hospitalized individuals.

**Post-acute symptoms stratified by age and sex**. Risk differences for symptoms 6–12 months after the test were stratified by age group and sex in order to assess the existence of subgroups at greater risk of post-acute symptoms (Fig. 4 and Supplementary Data 1). Based on descriptive results, the majority of post-acute symptoms tended to more often be reported by females and especially by 30–59-year-old participants. Stratified RDs for experiencing at least one of the symptoms: fatigue/exhaustion, dysgeusia, dysosmia, 6–12 months after test, were higher for females (RD = 18.0%, 95% CI 17.5–18.5%) compared to males (RD = 13.1%, 95% CI 12.6–13.5%). In addition, RDs for experiencing at least one of these symptoms were higher for 30–59 year olds (RD = 18.2%, 95% CI 17.7–18.7%) compared to for all other age groups (15–29 and 60 +) (RD = 13.5%, 95% CI 13.0–13.9%).

| Symptom | Positive (n,%) | Negative (n,%) | | RD (95% C.I.) |
|---|---|---|---|---|
| Dysosmia | 6,674 (10.9%) | 604 (0.7%) | | 10.92 (10.64, 11.20) |
| Dysgeusia | 5,365 (8.8%) | 551 (0.6%) | | 8.68 (8.43, 8.93) |
| Fatigue/exhaustion | 6,799 (11.1%) | 2,868 (3.1%) | | 8.43 (8.12, 8.74) |
| Dyspnea | 3,277 (5.4%) | 813 (0.9%) | | 4.87 (4.64, 5.07) |
| Reduced strength legs/arms | 3,381 (5.5%) | 1,024 (1.1%) | | 4.68 (4.45, 4.90) |
| Sleeping legs/arms | 2,841 (4.7%) | 1,236 (1.3%) | | 3.50 (3.30, 3.71) |
| Muscle/joint pain | 3,217 (5.3%) | 1,772 (1.9%) | | 3.46 (3.24, 3.68) |
| Headache | 3,740 (6.1%) | 2,868 (3.1%) | | 3.04 (2.79, 3.30) |
| Dizziness | 2,430 (4.0%) | 1,495 (1.6%) | | 2.38 (2.18, 2.58) |
| Chest pain | 1,695 (2.8%) | 780 (0.8%) | | 2.01 (1.85, 2.16) |
| Hot flushes/sweat | 2,047 (3.4%) | 1,550 (1.7%) | | 1.66 (1.48, 1.84) |
| Reduced appetite | 1,772 (2.9%) | 1,176 (1.3%) | | 1.51 (1.36, 1.67) |
| Red runny eyes | 822 (1.3%) | 748 (0.8%) | | 0.50 (0.38, 0.62) |
| Abdominal pain | 1,241 (2.0%) | 1,410 (1.5%) | | 0.44 (0.29, 0.60) |
| Chills | 966 (1.6%) | 980 (1.1%) | | 0.44 (0.30, 0.56) |
| Nausea | 1,179 (1.9%) | 1,294 (1.4%) | | 0.43 (0.28, 0.59) |
| Diarrhoea | 1,122 (1.8%) | 1,338 (1.5%) | | 0.34 (0.20, 0.51) |
| Fever | 1,362 (2.2%) | 1,584 (1.7%) | | 0.32 (0.16, 0.48) |
| Cough | 2,956 (4.8%) | 4,077 (4.4%) | | −0.01 (−0.23, 0.22) |
| Runny nose | 2,376 (3.9%) | 3,474 (3.8%) | | −0.22 (−0.43, −0.01) |
| Sore throat | 2,282 (3.7%) | 3,690 (4.0%) | | −0.65 (−0.85, −0.43) |

**Fig. 1 Risk differences of symptoms after 6–12 months, comparing SARS-CoV-2 test-positive and test-negative participants.** Note: Bars indicate risk differences (center) with 95% confidence intervals (length of error bars) adjusted for age, sex, comorbidities, obesity, healthcare occupation, and time after testing (in months). For post-acute symptoms 6–12 months after the test date, all test negatives no matter the indication for testing are used as the control population. All symptom questions were mandatory, so for all lines the proportions are based on 61,002 test-positive and 91,878 test-negative individuals. An individual could only participate in the study once, as either test-positive or test negative.

| Medical diagnosis | Positive (n,%) | Negative (n,%) | | RD (95% C.I.) |
|---|---|---|---|---|
| Chronic fatigue syndrome | 2,401 (4.0%) | 1,329 (1.5%) | | 2.53 (2.35, 2.71) |
| Anxiety | 1,900 (3.4%) | 1,783 (2.1%) | | 1.15 (0.95, 1.34) |
| Depression | 1,883 (3.5%) | 1,870 (2.3%) | | 1.00 (0.81, 1.19) |
| PTSD | 769 (1.3%) | 1,100 (1.2%) | | 0.16 (0.03, 0.28) |
| Fibromyalgia | 620 (1.0%) | 986 (1.1%) | | 0.02 (−0.09, 0.14) |

**Fig. 2 Risk differences of self-reported new diagnoses received between the test date and until 6-12 months after, comparing SARS-CoV-2 test-positive and test-negative participants.** Note: Bars indicate risk differences (center) with 95% confidence intervals (length of error bars) adjusted for age, sex, comorbidities, obesity, healthcare occupation and time after testing (in months). PTSD = post-traumatic stress disorder. For diagnoses with onset between the test date and until 6–12 months after the test date, all test negatives no matter of the indication for testing are used as control population. All symptom questions were mandatory, so for all lines the proportions are based on 61,002 test-positive and 91,878 test-negative individuals. An individual could only participate in the study once, as either test-positive or test negative.

| Health problem | Positive (n,%) | Negative (n,%) | | RD (95% C.I.) |
|---|---|---|---|---|
| Physical exhaustion | 25,492 (45.5%) | 5,879 (7.3%) | | 40.45 (39.99, 40.97) |
| Mental exhaustion | 20,810 (37.7%) | 5,877 (7.4%) | | 32.58 (32.11, 33.09) |
| Difficulties concentrating | 16,720 (29.7%) | 2,812 (3.4%) | | 28.34 (27.91, 28.78) |
| Memory issues | 16,149 (28.7%) | 3,057 (3.7%) | | 27.25 (26.80, 27.71) |
| Sleep problems | 11,850 (22.9%) | 4,936 (6.5%) | | 17.27 (16.81, 17.73) |

**Fig. 3 Risk differences of self-reported health problems with new onset between the test date and until 6–12 months after, comparing SARS-CoV-2 test-positive and test-negative participants.** Note: Bars indicate risk differences (center) with 95% confidence intervals (length of error bars) adjusted for age, sex, comorbidities, obesity, healthcare occupation, and time after testing (in months). For health problems with onset between the test date and until 6–12 months after the test date, all test negatives no matter of the indication for testing are used as control population. All symptom questions were mandatory, so for all lines the proportions are based on 61,002 test-positive and 91,878 test-negative individuals. An individual could only participate in the study once, as either test-positive or test negative.

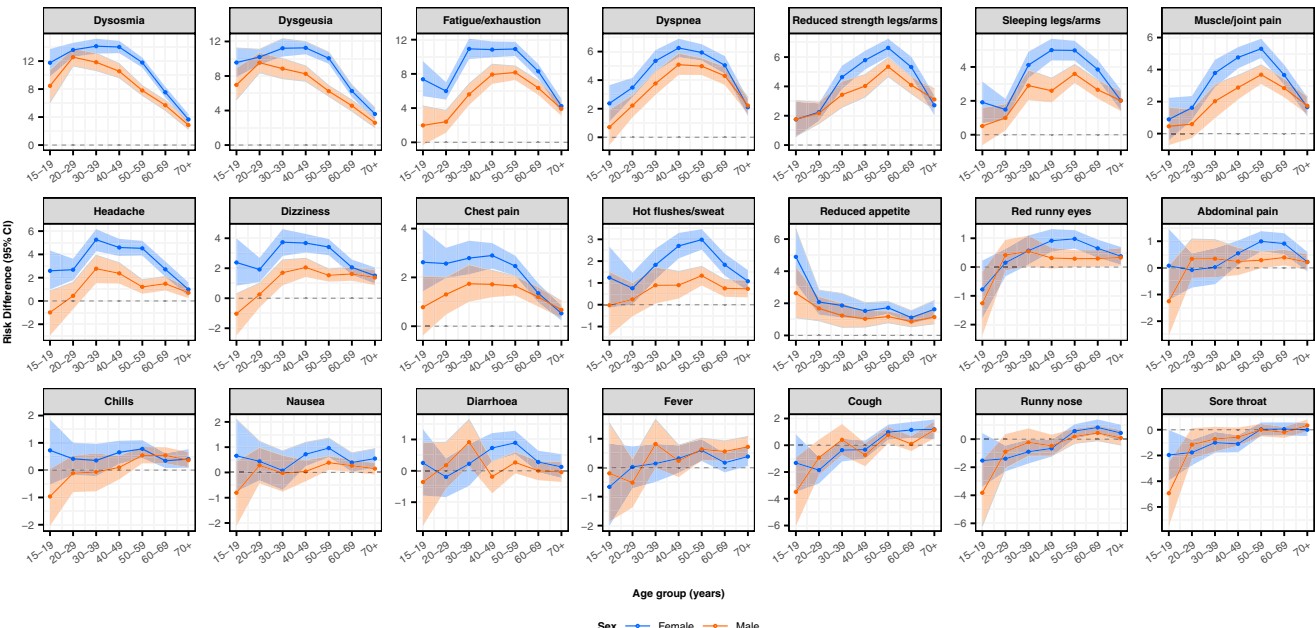

**Fig. 4 Risk differences of symptoms after 6-12 months, comparing SARS-CoV-2 test-positive and test-negative participants, stratified by sex and age group.** Note: Risk differences (center) with 95% confidence intervals (width of error bands) were adjusted for comorbidities, obesity, healthcare occupation, and time after testing (in months).

Similar trends and more pronounced differences were observed for new onset of memory-, concentration-, or sleep problems, as well as mental or physical exhaustion (Supplementary Fig. 3 and Supplementary Data 2). Risk differences for new onset of diagnoses of anxiety were highest among 20–29-year-old females (Supplementary Fig. 4 and Supplementary Data 3). Depression were more often reported by 30–39-year-olds regardless of sex.

**Sick leave**. Full or part-time sick leave was more common after a SARS-CoV-2 infection. Among the test positives 12.0% reported taking any sick leave 4 weeks after the test and until filling in the questionnaire 6–12 months later, compared to 7.7% of test negative (RD = 4.32%, 95% CI 4.00–4.64%). Full-time sick leave was reported by 9.4% of test positives and 6.5% of test negative (RD = 3.20, 95% CI 2.88–3.47%), whereas part-time sick leave was reported by 4.2% of test positives compared to 1.7% of test negative (RD = 2.43%, 95% CI 2.25–2.62%). Some individuals reported both full- and part-time sick leave.

## Discussion

Individuals testing positive for SARS-CoV-2 in Denmark during the study period and completing a questionnaire 6–12 months after the test date, more often reported post-acute symptoms, and new-onset diagnoses, and other health problems at any time since the test date, compared to test-negative individuals. In particular, there was a marked overrepresentation of self-reported physical and mental exhaustion among the test positives, as well as difficulties concentrating, memory issues, and sleep problems. New diagnoses of CFS, depression, and anxiety were also more common after testing positive. The highest risk differences for physical symptoms were observed for dysosmia, dysgeusia, fatigue/exhaustion, and dyspnea. This is consistent with other findings among mainly non-hospitalized patients[8–12].

Differences in included symptoms, varying follow-up times, methodology and lack of control groups make direct comparisons between studies difficult. Even for studies of self-reported symptoms in the general, mainly non-hospitalized population, the proportion of cases reporting at least one symptom at least three months after testing positive varies markedly from 2.3[13] to 37.7%[14], where the latter are comparable to the observed proportion in the present study (29.6%). Interestingly the proportion of test negative, who reported at least one symptom, also varies considerably between studies, e.g., 3%[14] compared to 13% in this study.

In a systematic review, the median prevalence of anosmia and dysgeusia were 11% (IQR, 5.7–14.3%, 19 studies) and 9% (IQR, 3.0–11.2%, 13 studies)[5], respectively, which is similar to in the present study. In a meta-analysis, the pooled proportion of individuals experiencing fatigue at least 12 weeks after diagnosis, was 32% (95% CI 27, 37)[15]. In this study, 11.1% of test positives reported fatigue/exhaustion within the past 14 days, when asked 6–12 months after test, whereas physical or mental exhaustion in general during the time since the test was reported by 45.5% and 37.7%, respectively. Generally, the reported symptom prevalences in our study are in the lower range compared to other studies. However, our study has longer follow-up time and is more representative of a general population where the majority of SARS-Cov-2-infected individuals have experienced milder disease. Thus, we believe that our study has greater external validity than many previous studies conducted in hospitalized- or otherwise selected populations.

It is well-established that neurocognitive sequelae in the form of anxiety, depression, cognitive problems and sleep disturbances may occur following COVID-19, but reported prevalences vary considerably[16]. Our results suggest that these problems are also prevalent among non-hospitalized individuals[17, 18]. The overrepresentation of CFS among test positives must be interpreted with care due to variability in how this diagnosis is made and the risk of confusing CFS with other conditions when filling in the questionnaire. However, increased incidence of CFS after COVID-19 have also been reported elsewhere[19].

The number of studies among non-hospitalized individuals with follow-up beyond 6 months are still limited. In one study including 794 test-positive individuals, no specific time gradient were observed in self-rated health 3–8 months post-infection[20]. Others have concluded that recovery beyond 6 months of illness was rare[21]. In the present study, a slightly decreasing trend in

reporting frequency at 6, 9, or 12 months was observed for most symptoms.

Increased frequency of post-acute symptoms in females compared to males and slower recovery in females have also been reported in other studies[8–10, 13, 14, 21, 22], whereas the evidence regarding the influence of age is somewhat contradicting. In one study, an inverted-U formed association between age and worsening of health after infection was observed, similar to this study, where the majority of symptoms were most frequently reported by the middle-aged (30–59 years)[20], but reports of increased risk in older individuals[9, 18], young adults[23], or no effect[8] also exist.

The reported differences in sick leave among test positives and test negatives indicate that post-acute symptoms are of such severity that they result in absence from work.

The main strengths of this study is its considerable size and the use of a large time-matched control population, making it possible to compare post-acute symptoms among COVID-19 cases and the background population represented by the control group. In addition, we were able to adjust for important confounders, including comorbidity. This allowed us to calculate adjusted risk difference measures for each acute and post-acute symptom, thus "deducting" the general morbidity in the population, including any general health effects that may have been caused by the lockdown or other societal restrictions put in place as part of the epidemic control.

The main limitations of the study are the self-reporting of symptoms and the participation rate. With little over 1/3 of the invitees choosing to participate, we cannot rule out participation bias. The motivation for participation could be higher among those experiencing post-acute symptoms, but on the other hand, those with very severe symptoms might not have had the energy to participate. Still, response rates among test positives and -negatives were similar. However, because of the size of the study and the marked risk differences between the case- and control groups, we believe that our results are valid.

In other to minimize the potential influence of recall bias on the reporting of post-acute symptoms, only symptoms experienced within the 14 days up to filling in the questionnaire, were included. For diagnoses made by a doctor or more general problems, we included the entire period since the test date. Thus, for general health problems occurring between the test date and completion of the questionnaire, we cannot rule out the possibility that some individuals have included problems occurring only during the acute phase. For neurocognitive problems, it is a limitation of the study that mainly diagnoses, and not a wider range of individual symptoms, have been included.

This study is focused on self-reported symptoms/disorders and does not use register data for study outcome definitions. In a recent study based on the Danish prescription, patient, and health insurance registers, it was found that compared with test-negative individuals, non-hospitalized SARS-CoV-2 test-positive individuals were at increased risk of being diagnosed with dyspnea and venous thromboembolism, but not other diagnoses[24]. To get the full picture of long-COVID, both types of studies are of importance.

We have not included information on vaccination in our study. By the end of the study period, only 6.8% of the Danish population had been fully vaccinated[25] (primarily those aged 85 years or older, individuals living in care homes, and frontline healthcare workers[26]).

The burden of self-reported symptoms, diagnoses, and health issues after SARS-CoV-2 infection appears to be significant in the Danish population and we believe the results are generalizable to other comparable populations. This should be taken into account, when evaluating the full impact of the pandemic and when evaluating the benefits of public health interventions aimed at preventing the spread of the virus.

Further research is needed to better understand, who is at increased risk of developing post-acute disease. Models for predicting post-acute disease based on acute symptoms during the first week have been developed[13], however, more information in particular on how post-acute disease can be prevented or treated is still needed. Furthermore, ongoing longitudinal studies are needed to provide more details, particularly on sustained mental health, fatigue, and cognitive problems, which this study found to be significantly more often reported among former COVID-19 patients than controls.

## Methods

**Study design and population**. In this nationwide cross-sectional survey, data on self-reported symptoms were collected using web-based questionnaires distributed via the national "e-Boks" system, which is a platform offering electronic postal communication with public authorities and the private sector (www.digst.dk). This system is used by 92% of all residents in Denmark aged 15 years and above.

In Denmark, unlimited access to reverse transcription-polymerase chain reaction (RT-PCR) tests for SARS-CoV-2 has been available for all adults since May 2020 independent of test indication in a so-called community test track[27]. All tests were free-of-charge and could be booked using an online booking system. Mass testing played a major role in handling the pandemic in Denmark, and during the period, where study participants were tested (September 1, 2020–April 2, 2021), the weekly PCR test incidence in Denmark ranged from 4,386–35,213 tests per 100,000 inhabitants (mean: 13,212)[28].

Individuals invited to participate in the study were selected based on RT-PCR test results recorded in the national COVID-19 surveillance system at Statens Serum Institut, which captures the individual results of all RT-PCR tests performed (https://covid19.ssi.dk/). All individuals who tested positive during September 1, 2020 to April 2 2021, and who had an e-Boks account were invited to participate, along with controls in the form of individuals testing negative only during the same period. Controls were randomly selected using incidence density sampling on the test date with a ratio of 2:3 between test positives and -negatives. This ratio was chosen to counteract a possible lower response rate among controls than in cases. Individuals receiving more than one positive test result during the study period, were included based on the first result, and an individual could only participate once as either case or control. The wild type (until end of 2020) and later Alpha were the predominant variants circulating in Denmark, during the period where participants in this study were infected[29]. Data were collected from August 1, 2021 to December 11, 2021, where participants received an invitation letter containing a link to the questionnaire 6, 9, or 12 months after their test date. Non-responders received a reminder 7–10 days after the invitation. The questionnaires were automatically locked 39–45 days after the invitation had been sent.

In order to minimize recall bias for acute symptoms, individuals with tests older than 12 months were not invited.

To avoid misclassification bias, controls who reported having been found seropositive were excluded. Participants were specifically asked to report any symptom that they might have experienced, no matter the reason, in order to avoid information bias from test positives omitting non-COVID-19 symptoms.

**Data sources**. Data were collected using questionnaires created in SurveyXact (www.surveyxact.dk), which could be completed using a PC, smartphone, or tablet. The questionnaire included questions on height, weight, education, employment, smoking and drinking habits, physical activity, sick leave, and symptoms in the time around the test date, defined as from 1 week before the test and until 4 weeks after. To evaluate post-acute COVID-19 symptoms, participants were asked about: (1) symptoms during the past 14 days, (2) selected health conditions diagnosed by a medical doctor before and after the test date, and (3) self-reported experiences of specific physical and neurocognitive symptoms 6 months before and up to 6–12 months after testing. For the reported symptoms and health conditions, participants were also asked about whether they used to regularly experience these before the test. Test negatives were asked about test indication and whether they suspected ever having had COVID-19. All questions in the questionnaire were mandatory, except height, weight, smoking, and alcohol consumption. The questionnaire is available as supplementary material (Supplementary Note 1).

In Denmark, individual-level data from different data sources can be linked using a unique identifier (the CPR-number) assigned in the Civil Registration System. Using the CPR-number, questionnaire data were supplemented with register-based information on age and sex, information on healthcare occupation from authorization data[30] as well as information on comorbidities and hospitalizations from the Danish National Patient Register (DNPR)[31]. The DNPR contains information on in- and outpatient diagnoses coded using ICD-10, which made it possible to calculate Charlson Comorbidity Index scores. Hospitalizations were considered COVID-19 related, if the patient had received a positive test result within 14 days of admission, and had been registered with one of the ICD-codes: DB342, DB342A, DB972, DB972A, DB972B, DB972B1, or DB948A. Hospital-acquired infections with SARS-CoV-2 were not included.

**Statistical methods**. The prevalence of conditions among test-positive and -negative individuals were compared using risk differences (RDs). Parametric g-computation[32] on logistic regression was used to estimate RDs (with 95% confidence intervals) among the exposed with adjustment for completion time (6, 9, or 12 months), age, sex, obesity, comorbidities from the questionnaire, Charlson Comorbidity Index scores, and healthcare occupation. Based on results from other studies[8–10, 20–23], these variables were considered potential confounders. Symptoms prior to the test were also adjusted for. For diagnoses and health conditions, only new onsets, defined as conditions occurring between testing and completion of the questionnaire, but not in the 6 months leading up to, were taken into account.

The 95% confidence intervals were obtained through bootstrap random resampling with 1000 iterations. The R-packages "riskCommunicator"[33] (v1.0.1) and "Forester" (v 0.5.0) were used for modeling and generation of forest plots, respectively. We estimated RDs for the following conditions: (1) acute symptoms in relation to the test date (only test negatives, who reported symptoms compatible with COVID-19 as indication for testing, were included as test negative in this analysis), (2) post-acute symptoms during the 14 days prior to questionnaire completion 6, 9, or 12 months after the test, (3) new onset diagnoses of anxiety, chronic fatigue syndrome, depression, fibromyalgia and post-traumatic stress disorder (PTSD) confirmed by a medical doctor since the test (onset between time of testing and questionnaire completion), and (4) new onset of mental or physical exhaustion, concentration difficulties, memory issues or sleep problems since the test (onset between the time of testing and questionnaire completion).

Main analyses were based on pooled data from 6, 9, or 12 months after tests and did not take time into account. Supplementary analyses were carried out at each of the three time points to examine if effects change time.

Charlson Comorbidity Index scores[34] were calculated based on data for the past 5 years extracted from the DNPR[31]. Scores were included in analyses as 0, 1 or ≥2, since very few had scores above 2. In the questionnaire, participants were asked supplementary questions about relevant comorbidities commonly treated in primary care (Supplementary Table 2) and therefore unlikely to be listed in the DNPR. Presence of these comorbidities were included in analyses as dichotomous variables. Obesity was defined as BMI ≥ 30 for individuals aged 18 years or above and for 15–17 years old international cut-off points for obesity by sex and age were used[35]. The distribution between groups for all variables adjusted for in analysis are listed in Supplementary Table 2.

P values in Table 1, Supplementary Tables 1, and 2 were estimated using student's t test for continuous variables and Pearson's Chi-squared test for categorical variables.

Data management and statistical analyses were conducted using R version 4.0.2[36].

**Ethical approval**. This study was performed as a surveillance study as part of the governmental institution Statens Serum Institut's (SSI) advisory tasks for the Danish Ministry of Health. SSI's purpose is to monitor and fight the spread of disease in accordance with section 222 of the Danish Health Act. According to Danish law national surveillance activities conducted by SSI does not require approval from an ethics committee. It was approved by the Danish Governmental law firm and SSI's compliance department that the study is fully compliant with all legal, ethical, and IT-security requirements and there are no further approval procedures regarding such studies.

Participation in the study was voluntary. The invitation letter to participants contained information about their rights under the Danish General Data Protection Regulation (rights to access data, rectification, deletion, restriction of processing and objection). It was considered informed consent, if potential participants after having read this information decided to click on the link in the invitation and fill in the questionnaire.

**Reporting summary**. Further information on research design is available in the Nature Research Reporting Summary linked to this article.

## Data availability

The datasets used in the study comprises individual-level sensitive information from completed questionnaires and national register data. According to the Danish data protection legislation, the authors are not allowed to share these sensitive data directly upon request. However, the data are available for research upon reasonable request to The Danish Health Data Authority (register-data, e-mail: kontakt@sundhedsdata.dk) and Statens Serum Institut (questionnaire data, e-mail: aii@ssi.dk) and within the framework of the Danish data protection legislation and any required permission from Authorities. Expect a time frame of at least 3–6 months for data requests to be processed.

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

## Acknowledgements
The authors would like to thank all EFTER-COVID participants for completing the questionnaire. We also thank, the members of the EFTER-COVID stakeholder group, consisting of representatives from universities, long-term sequelae clinics and other relevant hospital departments or health institutions, for their comments to the questionnaires and input during meetings. No specific funding was received for this work that was conducted as part of the governmental institution Statens Serum Institut's advisory tasks for the Danish Ministry of Health.

## Author contributions
The study was designed and initiated by P.B., S.E., A.H., N.N., A.K., and A.S. The questionnaire was designed by A.S., P.B., N.N., A.K., S.E., and I.M. Collection of the data including programming related to this: A.S., J.H., P.B., and I.M. Data analysis was done by L.S. The first draft was written by A.S., L.S., and A.H. All authors have critically revised the manuscript. A.S. is the guarantor of the overall content and accepts full responsibility for the work. The data underlying this study were accessed and verified by L.S. and A.S. All authors have approved the final version of the manuscript. The corresponding author confirms that all listed authors meet authorship criteria and that no one meeting the criteria have been omitted.

## Competing interests
The authors declare no competing interests.
