## [Peer Review File · Nature Communications]

A nationwide Danish questionnaire study of post-acute symptoms and health problems after SARS-CoV-2 infectionREVIEWER COMMENTS

Reviewer #1 (Remarks to the Author):

Interesting analysis but suffers from many serious flaws.

I have several concerns for your consideration.

a. the use of negative test design is fraught with bias. People are not universally tested, and the test is not randomly administered. People get tested for an indication: a) known exposure or symptoms suggestive of covid-19 or b) need to test for another reason including need to interact with a healthcare system. For example, cancer patients in need of a biopsy get tested before undergoing the procedure, and the results in these patients are often negative. If the investigators examine it, they will find that compared to covid-19 positive test, there is an association between a negative test and cancer, chemotherapy, and diagnostic procedures for cancer. Clearly this is due to a bias related to cancer patients needing to get a covid-19 test before interacting with the health care system. There are million example like this one. Please reconsider the design and include a control group of people with no known infection (instead of the negative test design).

b. I was struck by the stark differences between the results presented in this manuscript and the findings -- using the same database by some of the same authors -- in the Lancet infectious diseases ([https://www.thelancet.com/action/showPdf?pii=S1473-3099\(21\)00211-5](https://www.thelancet.com/action/showPdf?pii=S1473-3099(21)00211-5)). That study found (among other things) no association between COVID-19 and fatigue (Figure 4 in that study), and was used by gaslighters to debase and discredit the existence of long covid. In my opinion, that study was biased too and illustrates the pitfalls of negative test design in that it was not able to reproduce the most well recognized association (covid-19 and fatigue). How do the findings in that study reconcile with the current analysis? I certainly realize that the follow up period is different, but this deserves discussion and more clarity. An analysis of the risk of these sequelae at different time points 60, 90, 180, 360 days would be helpful.

c. The authors should test positive outcome and negative outcome controls. and also consider testing negative exposure controls to help assuage concerns about spurious biases.

d. Another major limitations is that the set of outcomes seem to be limited to data from the questionnaire. and it is likely to miss a lot of the important sequelae of long covid (for example, heart disease, neurologic disorders, diabetes, kidney disease, etc...).

Reviewer #2 (Remarks to the Author):

see attached, uploaded file.

The results are not new but consistent with previous published work in this field (as quoted in their references) and add confidence because of the size and rigour of the study and its national coverage. The 1/3 response rate is higher than for many questionnaire studies.

The results section supports the discussion and conclusions.

All weaknesses and caveats to the interpretation and conclusions are described in an open and balanced way.

The methodology is sound and of a high standard.

There is enough detail provided in the methods for the work to be reproduced.

Reviewer #3 (Remarks to the Author):

This is a very important question as there have to date been only a few case-control studies such as the UK REACT 2 study and the literature is predominantly on symptom self report uncontrolled cohorts or case sees. As such this is a potentially very useful paper and is timely.

Important positive points

- Free RT-PCR testing without needing a specific indication in Denmark.
- +ve test status confirmed in national database and free of reporting bias.
- Contemporaneous Test-ve controls (excluding any seropositives)

Potential weak points

- Potential for bias in retrospective recall.
- Low response rate (35.7%) - but reasonable for this type of research.

Suggestions to improve the paper

1. It would be worth pointing out that the questionnaire had a long list of symptoms (see appendix) and the possibility to include free text. I note that post-exertional malaise/symptom exacerbation is not on the list. In addition the questionnaire elicits responses over a period of up to six months from the +ve test, which could have been up to 12 months previously.
2. The neurocognitive symptoms are dealt with separately in a six month before /six months after dichotomy- In other studies (esp self report) these are some o the most bothersome (brain fog etc) and it is very strange to see them missed out from the main list. This omission - due to the way the data was collected should be noted.
3. It should also be noted that 'anxiety symptoms' doesn't necessarily mean anxiety as a diagnosis. As tachycardia, sleeplessness and a feeling of anxiety can be caused by dysautonomia, which is frequently reported in Long Covid. I don't think these should be described in the discussion as 'psychiatric' as that implies an aetiology. Please use 'neurocognitive' instead.
4. I think there is a potential missing aspect to the analysis. We are shown risk-differences between cases and controls. What I would liked to see are the odds ratios for significant predictors of self reported Long Covid to tease out what a data-based symptom definition looks like. Unfortunately the subjects were not asked if they considered themselves to have 'Long Covid', so there is nothing to 'predict' in a model. I wonder if the question 'is your health returned to normal' could be used as a proxy for this? Given that at present Long Covid has been patient-defined and lacks a specific biomarker (s) its perfectly reasonable to ask what symptoms independently predict a patient feeling that they are not 'back to normal'.
5. Regardless the issues of age and sex, deprivation etc could be dealt with using regression models to predict 'at least one symptom'.
6. The REACT 2 study has now been published. Worth making a comparison in the discussion. <https://www.nature.com/articles/s41467-022-29521-z>. Similar results by the look of it (up to 12 weeks), but REACT 2 adds a cluster analysis of symptoms.

On balance the paper is a sound, useful and timely addition to the literature. There are some things that can be done to improve clarity and I would like the authors to consider some additional analysis to make comparisons easier with other papers.

Reviewer #4 (Remarks to the Author):

This is an interesting study of long covid in the general population including 153k people. The authors investigate the acute and post acute symptoms as well as post acute symptoms duration. The paper is well written and topic is highly relevant. However, I have some concerns

1. I found numbers and inclusion/exclusion criteria a bit confusing. In the results section, the authors state that 153,412 invited participants fully completed the questionnaire, but two lines below they talk about 61,002 cases and 91,878 controls (total 152,880). The missing 532 individuals had apparently a positive serology test. A consort diagram indicating inclusion and exclusion criteria and the numbers of included individuals at each step would be very useful
2. Also, did all the test negative individuals have a serology test? If not, how can the authors know whether the negative controls were negative all through the 12 months follow-up period?
3. As the study is mainly focusing on long covid, it would be good to highlight this more and provide an estimate on how many people can be classified as suffering from long covid
4. are the RDs reported in the paper all adjusted for covariates? Are they also adjusted for multiple testing.
5. The discussion would benefit from a more thorough review of the literature including 1. PMID: 33692530, PMID: 34209085, PMID: 35429399
6. the authors should specify, if possible, to what variant do their results relate

Reviewer #2 (Remarks to the Author):

Section	Page	Text from manuscript	My comments
Abstract	2		Excellent abstract.
Main (Intro)	2-3		Excellent. It would be better English to write "A nationwide questionnaire study was conducted" ...
Results	Page 3, lines 91 and 103	Among the 171,992 test-positives and 258,181 test-negatives, who were invited to participate,	Why were the test negatives tested? Did they all need symptoms to obtain a test or were some carried out through screening of contacts or workplace related? Line 251 of the Methods states that there were no selection criteria in Denmark to obtain a PCR test between sept 2020-April 2021 which was freely available to all.
	Page 3, line 104	Among all test negatives, irrespective of test indication, 13.5% reported at least one symptom around the test date with a median of two different symptoms.	This seems to aggregate all the test negatives, irrespective of whether they were tested because of symptoms or because of asymptomatic screening
	Fig 1		Demonstrates very nicely that the classic symptoms of coryza associated with a viral respiratory tract infection or viral acute gastroenteritis all subside by 6 to 12 months whereas the persisting symptoms are those which other authors have described in relation to long Covid.
Methods - Cohorts	8 line 256	controls in the form of individuals testing negative only during the same period.	Why were the test negatives tested? Did they all need symptoms to obtain a test or were some carried out through screening of contacts or workplace related? Lines 278-279 state "Test-negatives were asked about test indication and whether they suspected ever having had COVID-19."
	9	dysgeusia	Not a term commonly used in English. ? loss/altered taste
Discussion	6	Generally, the reported symptom prevalences in our study are in the lower range compared to other studies. However, our study has longer follow-up time and is more	Correct. The Strengths & Limitations section is well written and balanced. I agree these are important points of difference.

		representative of a general population where the majority of individuals have experienced milder disease. Thus, we believe that our study has greater external validity than many previous studies conducted in hospitalized- or otherwise selected populations.	
	7	The main strengths of the present study is its considerable size and the use of a large time-matched control population, making it possible to compare post-acute symptoms among COVID-19 cases and the background population represented by the control group.	I agree
	7	With little over 1/3 of the invitees choosing to participate, we cannot rule out participation bias. The motivation for participation could be higher among those experiencing post-acute symptoms, but on the other hand, those with very severe symptoms might not have had the energy to participate. Still, response rates among test-positives and –negatives were similar.	I agree

REVIEWER COMMENTS

Reviewer #1 (Remarks to the Author):

Interesting analysis but suffers from many serious flaws.
I have several concerns for your consideration.

a. the use of negative test design is fraught with bias. People are not universally tested, and the test is not randomly administered. People get tested for an indication: a) known exposure or symptoms suggestive of covid-19 or b) need to test for another reason including need to interact with a healthcare system. For example, cancer patients in need of a biopsy get tested before undergoing the procedure, and the results in these patients are often negative. If the investigators examine it, they will find that compared to covid-19 positive test, there is an association between a negative test and cancer, chemotherapy, and diagnostic procedures for cancer. Clearly this is due to a bias related to cancer patients needing to get a covid-19 test before interacting with the health care system. There are million example like this one. Please reconsider the design and include a control group of people with no known infection (instead of the negative test design).

We understand that this might be the situation in many countries and that the manuscript could have benefitted from including a thorough explanation of the Danish Test system. A paragraph about this have now been added in l. 270-9.

Throughout the major part of the pandemic, Denmark has had one of the highest test incidences in Europe (<https://www.ecdc.europa.eu/en/publications-data/covid-19-testing>) as well as globally (Hasell et al, 2020, <https://ourworldindata.org/coronavirus-testing>). During May 2020, PCR-tests became freely available to all citizens, independent of test indication, in the so-called “community test track”, where the capacity was gradually upscaled to ~ 170,000 daily tests (DK population: 5.8 mio.) towards the end of the present study

period April 2021, (<https://tcdk.ssi.dk/om-testcenter-danmark/testkapacitet-gennem-tiden>). The “community test track” was running in parallel with the “health care test track”, where persons admitted to hospital were tested. All tests were free-of-charge PCR and could be booked using an online booking system. In many periods walk-in tests was also available at some test sites.

During the period, where study participants were tested (September 1, 2020 – April 2, 2021), the weekly PCR test incidence in DK ranged from 4,386-35,213 tests per 100,000 inhabitants (mean: 13,212) (Source: Data from <https://www.ecdc.europa.eu/en/publications-data/covid-19-testing>). When booking a PCR test in the “community test track” online, the person booking was invited to answer a few questions about the reason for booking. During the period relevant for this study, among the 6,319,282 replies to these questions, the top-5 reported reasons for booking were: 1) I need to get tested in relation to work or school/education (20.4%), 2) Other reason / participation in research (19.6%), 3) I have been in contact with an infected person (19.4%), 4) I have or have had covid-19 like symptoms (11.5%), and 5) I want to get tested prior to participating in an event (e.g. sports event, wedding) (10.2%) (Statens Serum Institute, unpublished data). The proportion of participants indicating planned hospital admission, dentist or other health care appointment as test indication was only 3.3% (Statens Serum Institut, unpublished data). Persons, who needed immediate hospital admission were tested in the “healthcare track” and thus are not included in the data for test indication presented here, but since these will constitute a rather limited proportion of the persons tested, we have no reason to believe that our study population are markedly biased by persons with cancer or other health conditions requiring regular medical attention.

b. I was struck by the stark differences between the results presented in this manuscript and the findings -- using the same database by some of the same authors -- in the Lancet infectious diseases ([https://www.thelancet.com/action/showPdf?pii=S1473-3099\(21\)00211-5](https://www.thelancet.com/action/showPdf?pii=S1473-3099(21)00211-5)). That study found (among other things) no association between COVID-19 and fatigue (Figure 4 in that study), and was used by gaslighters to debase and discredit the existence of long covid. In my opinion, that study was biased too and illustrates the pitfalls of negative test design in that it was not able to reproduce the most well recognized association (covid-19 and fatigue). How do the findings in that study reconcile with the current analysis? I certainly realize that the follow up period is different, but this deserves discussion and more clarity. An analysis of the risk of these sequelae at different time points 60, 90, 180, 360 days would be helpful.

The study in Lancet Infectious Diseases is a register-based study, where the objective is to detect changes in hospital-acquired diagnoses, prescription patterns and frequency of seeking medical attention following a covid diagnosis e.g. GP contacts. Thus, this setup is well-suited for detecting rare, but severe long-term effects which might require hospital treatment and medication, and not so much for detecting more general long-term health problems, such as fatigue, since individuals experiencing this type of problems might only be seen by their general practitioner (GP), who only reports minimal clinical information to the Danish National Health Insurance Register, and not referred to hospital within the follow-up period used in that study. Therefore this study and the present study based on self-reported symptoms supplement each other nicely, since the present study will be able to detect more frequent reporting of general symptoms among test-positives, which could still substantially impact quality of life. Additionally, fatigue could potentially be one of the causes of the increased number of GP visits observed among test-positives in the register-based study in Lancet Infectious Diseases.

We agree that it would be valuable with an analysis of the risk at different time points. In a secondary analysis, we do evaluate risk differences at the 180, 270 and 360 day time points, “When looking at estimated RDs for questionnaires completed at 6, 9 or 12 months separately, RDs tended to decrease over

time. Among the ten symptoms with largest overall RDs, the estimates decreased over time for all except dysosmia and dysgeusia for which estimates were largest at 9 months (table S3).” (l. 144-6). We find that self-reported fatigue/exhaustion is still significantly increased at the 360 day time point, risk difference 6.95 (95% CI, 6.24 to 7.77) (Table S3).

c. The authors should test positive outcome and negative outcome controls. and also consider testing negative exposure controls to help assuage concerns about spurious biases.

We address above why selection bias is not a concern given the widespread community testing in Denmark. If controls were selected from the untested population, these might differ from the cases in several ways e.g. in terms of socioeconomic status, occupation and social activity, and in order to obtain the most accurate picture of whether ongoing symptoms are more common among test-positives than test-negatives, we believe it will reduce bias if the two groups compared are as similar as possible, when it comes to all other covariates, than the one of interest (test-status).

d. Another major limitations is that the set of outcomes seem to be limited to data from the questionnaire. and it is likely to miss a lot of the important sequelae of long covid (for example, heart disease, neurologic disorders, diabetes, kidney disease, etc...).

We agree that these are important outcomes to study. However, these outcomes are best evaluated in register-based studies using diagnoses codes. Studies focusing on the rarer and most severe symptoms resulting in hospital contact and/or prescription of medicine fail to take the less specific symptoms affecting the majority of cases into account. This is best evaluated in a questionnaire survey such as ours. Both types of studies are warranted in order to get the full picture of the long term effects of covid. We now discuss this issue in more detail (l. 244-8).

Reviewer #2 (Remarks to the Author):

see attached, uploaded file (has been copied in below and an “author response column” has been added)

Section	Page	Text from manuscript	My comments (reviewers)	Response from authors
Abstract	2		Excellent abstract.	Thanks
Main (Intro)	2-3		Excellent. It would be better English to write “A nationwide questionnaire study was conducted” ...	Thanks, the wording has now been updated

Results	Page 3, lines 91 and 103	Among the 171,992 test-positives and 258,181 test-negatives, who were invited to participate,	Why were the test negatives tested? Did they all need symptoms to obtain a test or were some carried out through screening of contacts or workplace related? Line 251 of the Methods states that there were no selection criteria in Denmark to obtain a PCR test between sept 2020-April 2021 which was freely available to all.	There was indeed no selection criteria during this period in Denmark. Based on the replies to a brief questionnaire, which persons were invited to fill-out, when booking a test, the top5 reasons for booking a PCR-test during this period was (N=6,319,282): 1) I need to get tested in relation to work or school/education (20.4%), 2) Other reason / participation in research (19.6%), 3) I have been in contact with an infected person (19.4%), 4) I want to get tested prior to participating in an event (e.g. sports event, wedding) (10.2%), and 5) I suspect that I might have been infected, but is not close contact or other contact to an infected (e.g. suspicion of having been infected during an event or larger gathering) (4.5%) (Statens Serum Institute, unpublished data). Clarification has been added in the methods section (l. 270-9)
	Page 3, line 104	Among all test negatives, irrespective of test indication, 13.5% reported at least one symptom around the test date with a median of two different symptoms.	This seems to aggregate all the test negatives, irrespective of whether they were tested because of symptoms or because of asymptomatic screening	Yes, this is correct – all test-negatives are included here irrespective of test indication. We asked both test-negatives and test-positives to mention all symptoms they experienced, no matter whether they thought that these were covid-related,

				hence asymptotically infected could also have mentioned symptoms caused by existing conditions (risk differences are estimated in order to take into account, that there will be a background level of some of these very general symptoms in the population)
	Fig 1		Demonstrates very nicely that the classic symptoms of coryza associated with a viral respiratory tract infection or viral acute gastroenteritis all subside by 6 to 12 months whereas the persisting symptoms are those which other authors have described in relation to long Covid.	Yes, we agree.
Methods - Cohorts	8 line 256	controls in the form of individuals testing negative only during the same period.	Why were the test negatives tested? Did they all need symptoms to obtain a test or were some carried out through screening of contacts or workplace related? Lines 278-279 state "Test-negatives were asked about test indication and whether they suspected ever having had COVID-19."	Please, see reply to reviewer comment re. page 3, lines 91 and 103, above.
	9	dysgeusia	Not a term commonly used in English. ? loss/altered taste	Both dysosmia and dysgeusia is commonly used in relation to covid in the medical literature. We do already provide explanations of these terms in non-medical terminology (l. 111-2).

Discussion	6	Generally, the reported symptom prevalences in our study are in the lower range compared to other studies. However, our study has longer follow-up time and is more representative of a general population where the majority of individuals have experienced milder disease. Thus, we believe that our study has greater external validity than many previous studies conducted in hospitalized- or otherwise selected populations.	Correct. The Strengths & Limitations section is well written and balanced. I agree these are important points of difference.	Thank you for this kind comment.
	7	The main strengths of the present study is its considerable size and the use of a large time-matched control population, making it possible to compare post-acute symptoms among COVID-19 cases and the background population represented by the control group.	I agree	Thank you.
		With little over 1/3 of the invitees choosing to participate, we cannot rule out participation bias. The motivation for participation could be higher among those experiencing post-acute symptoms, but on the other hand, those with very severe symptoms might not have had the energy to participate. Still, response rates among test-positives and –negatives were similar.	I agree	Thank you.

The results are not new but consistent with previous published work in this field (as quoted in their references) and add confidence because of the size and rigour of the study and its national coverage. The 1/3 response rate is higher than for many questionnaire studies.

The results section supports the discussion and conclusions.

All weaknesses and caveats to the interpretation and conclusions are described in an open and balanced way.

The methodology is sound and of a high standard.

There is enough detail provided in the methods for the work to be reproduced.

Thank you for the encouraging comments about our study.

Reviewer #3 (Remarks to the Author):

This is a very important question as there have to date been only a few case-control studies such as the UK REACT 2 study and the literature is predominantly on symptom self report uncontrolled cohorts or case sees. As such this is a potentially very useful paper and is timely.

Response from the authors: Thank you for the nice comments.

Important positive points

- Free RT-PCR testing without needing a specific indication in Denmark.
- +ve test status confirmed in national database and free of reporting bias.
- Contemporaneous Test-ve controls (excluding any seropositives)

Potential weak points

- Potential for bias in retrospective recall.
- Low response rate (35.7%) - but reasonable for this type of research.

Yes, we agree.

Suggestions to improve the paper

1. It would be worth pointing out that the questionnaire had a long list of symptoms (see appendix) and the possibility to include free text. I note that post-exertional malaise/symptom exacerbation is not on the list. In addition the questionnaire elicits responses over a period of up to six months from the +ve test, which could have been up to 12 months previously.

We agree that it might also have been relevant to include questions related to PEM. However, in order to keep the questionnaire at a reasonable length, we had to prioritize. The results presented in this manuscript will be supplemented by results from an ongoing longitudinal study, where we have several parallel “tracks”, each focusing on a special aspect of long-COVID. One of the tracks will focus on fatigue and includes PEM-items adopted from the DePaul Symptom Questionnaire (Cotler et al., 2018).

Regarding the duration between testing positive or negative and completing the questionnaire, participants received the invitation to fill out the questionnaire, exactly 6, 9 or 12 months after the test date of reference, meaning that new questionnaires were distributed every day during the relevant period. If invited persons had not accessed the questionnaire within 39-45 days, they would be considered non-responders and the questionnaire would be locked (for technical reasons the interval varied depending on which day of the week the questionnaire was received). Therefore all questionnaires will have been completed within the stated time + a maximum of 45 days. A sentence has been added in the methods section (l. 292-3) in order to make this clear.

2. The neurocognitive symptoms are dealt with separately in a six month before /six months after dichotomy- In other studies (esp self report) these are some of the most bothersome (brain fog etc) and it is very strange to see them missed out from the main list. This omission - due to the way the data was collected should be noted.

To keep the questionnaire at a reasonable length – in order to maximise the participation rate - we had to prioritise and we chose to focus on the physical symptoms in our current study and the neurocognitive symptoms in our ongoing longitudinal study. We believe that this approach will provide detailed and more accurate information on both physical symptoms and neurocognitive symptoms, and, in the end, give a more comprehensive picture of long covid.

We have now inserted a sentence in the discussion to highlight that the lack of detail re. neurocognitive symptoms is a limitation of the study (l. 241-3)

3. It should also be noted that ‘anxiety symptoms’ doesn’t necessarily mean anxiety as a diagnosis. As tachycardia, sleeplessness and a feeling of anxiety can be caused by dysautonomia, which is frequently reported in Long Covid. I don’t think these should be described in the discussion as ‘psychiatric’ as that implies an aetiology. Please use ‘neurocognitive’ instead.

Thank you for pointing this out. ‘Psychiatric’ has now been replaced by ‘neurocognitive’.

4. I think there is a potential missing aspect to the analysis. We are shown risk-differences between cases and controls. What I would like to see are the odds ratios for significant predictors of self reported Long Covid to tease out what a data-based symptom definition looks like. Unfortunately the subjects were not asked if they considered themselves to have ‘Long Covid’, so there is nothing to ‘predict’ in a model. I wonder if the question ‘is your health returned to normal’ could be used as a proxy for this? Given that at present Long Covid has been patient-defined and lacks a specific biomarker (s) its perfectly reasonable to ask what symptoms independently predict a patient feeling that they are not ‘back to normal’.

Thank you for the suggestion, this would indeed be an interesting analysis. Unfortunately the wording of the question in our questionnaire you are referring to is “Did your health return to normal within 4 weeks of testing positive for COVID-19?”. Even if participants reply “No” to this, they do not full-fill the WHO case definition for long-COVID (which unfortunately was published after this study had already started). Among the test-positive study participants 55.0% replied “No” to their health having returned to normal at this stage (53.8% among the non-hospitalised majority, and 80.7% among those being hospitalised due to covid (but not receiving ventilator treatment), and 90.1% among those put on ventilation).

5. Regardless the issues of age and sex, deprivation etc could be dealt with using regression models to predict 'at least one symptom'.

The goal of our study was not to construct a prediction model for long covid, but to describe and identify post-acute symptoms associated with infection. We do not believe a prediction model is the optimal approach to evaluating associations between infection and post-acute symptoms. Furthermore, the symptoms in our study are not uncommon (e.g. 13% of test-negatives report at least one symptom) and the interpretation and usefulness of a prediction model for "at least one symptom" is unclear.

6. The REACT 2 study has now been published. Worth making a comparison in the discussion. <https://www.nature.com/articles/s41467-022-29521-z>. Similar results by the look of it (up to 12 weeks), but REACT 2 adds a cluster analysis of symptoms.

Thank you for pointing us in the direction of this newly published paper. A comparison has now been made in the discussion (l. 188-93)

On balance the paper is a sound, useful and timely addition to the literature. There are some things that can be done to improve clarity and I would like the authors to consider some additional analysis to make comparisons easier with other papers.

Thank you. Comparisons to other studies will be challenging due to differences in follow-up time, test strategy in individual countries and lack of standardised long covid phenotypes.

Reviewer #4 (Remarks to the Author):

This is an interesting study of long covid in the general population including 153k people. The authors investigate the acute and post acute symptoms as well as post acute symptoms duration. The paper is well written and topic is highly relevant.

Thank you for the kind words.

However, I have some concerns

1. I found numbers and inclusion/exclusion criteria a bit confusing. In the results section, the authors state that 153,412 invited participants fully completed the questionnaire, but two lines below they talk about 61,002 cases and 91,878 controls (total 152,880). The missing 532 individuals had apparently a positive serology test. A consort diagram indicating inclusion and exclusion criteria and the numbers of included individuals at each step would be very useful

There is only one exclusion step – the exclusion of those how who reported having received a positive serology result. The information in l. 95-7 has now been moved up (l. 88-90) in order to make the existence

of this exclusion step more clear earlier on when reading through the results section.

2. Also, did all the test negative individuals have a serology test? If not, how can the authors know whether the negative controls were negative all through the 12 months follow-up period?

No, the participants did not have a serology test. We can in principle not know for sure that the control persons have never been infected, but we do know that they were not recorded with a positive PCR result before the date where they were invited to fill in the questionnaire (6, 9 and 12 months after the test date of reference). In Denmark, only PCR-tests were available during the study period, but the possibility of not getting testing or having been tested positive abroad of course exists. However, given the relatively high weekly PCR-test incidence in Denmark during the study period (4,386-35,213 tests per 100,000 inhabitants (mean: 13,212)), we do not believe that this potential source of error will have had any major impact on the results.

3. As the study is mainly focusing on long covid, it would be good to highlight this more and provide an estimate on how many people can be classified as suffering from long covid

We agree that it would be valuable to be able to present an estimate of how many people are affected by long-COVID. However, given that there is no well-defined long covid phenotype currently, and that the WHO case definition lacks specificity (13% of the controls person also report at least one symptom), we thought it more informative to report and highlight risk differences for individual symptoms.

4. are the RDs reported in the paper all adjusted for covariates? Are they also adjusted for multiple testing.

Yes, as described in l. 326-9 in the method section all RDs are adjusted for completion time (6, 9 or 12 months), age, sex, obesity, comorbidities from the questionnaire, Charlson Comorbidity Index scores based on registration in the Danish National Patient Registry and healthcare occupation. In this study, an individual could only be invited to participate once (has been added in l. 288). We do not adjust for multiple testing.

5. The discussion would benefit from a more thorough review of the literature including 1. PMID: 33692530, PMID: 34209085, PMID: 35429399

- The references PMID: 33692530 and PMID: 35429399 have now been included in the discussion (l. 188-91 + 257-59) and main (l. 65-67), respectively. PMID: 34209085 was conducted among hospitalized patients, and we have chosen to focus the discussion on studies where a more comparable study population have been used.

6. the authors should specify, if possible, to what variant do their results relate

During the period, where participants in this study were tested, the Wuhan (until end of 2020) and later Alpha were the pre-dominating variants circulating in Denmark. This information has now been added in l. 288-90.

REVIEWERS' COMMENTS

Reviewer #1 (Remarks to the Author):

The authors addressed my comments satisfactorily.

Reviewer #3 (Remarks to the Author):

I have no particular comments on the revised paper. The authors have responded carefully to my comments and introduced changes when required. I accept their explanation for not being able to conduct a model based on self-assessed Long Covid.

The comments of the other referees seem to have been dealt with fairly and thoroughly by the authors.

The paper is a useful addition to the literature.

Reviewer #4 (Remarks to the Author):

The authors have appropriately addressed my concerns, however I really believe they ought to adjust for multiple testing.

Dear Editor and reviewers,

Thank you again for the review and inputs for improvement of the manuscript.

Below, we respond to your comments point-to-point.

Sincerely,

Anna Irene Vedel Sørensen, BSc, MSc, PhD
Section for Zoonotic, Food and Waterborne Infections
Dept. of Infectious Disease Epidemiology & Prevention
Statens Serum Institut

REVIEWER COMMENTS

Reviewer #1 (Remarks to the Author):

The authors addressed my comments satisfactorily.

Thank you

Reviewer #3 (Remarks to the Author):

I have no particular comments on the revised paper. The authors have responded carefully to my comments and introduced changes when required. I accept their explanation for not being able to conduct a model based on self-assessed Long Covid.

the comments of the other referees seem to have been dealt with fairly and thoroughly by the authors.

the paper is a useful addition to the literature.

Thank you

Reviewer #4 (Remarks to the Author):

The authors have appropriately addressed my concerns, however I really believe they ought to adjust for multiple testing.

Thank you

If and how to adjust for multiple testing is debatable – see e.g. <https://pubmed.ncbi.nlm.nih.gov/2081237/>. We prefer not to adjust for multiple testing. Adjusting for multiple testing reduces type I errors but at the cost of increasing type II errors. Since our study is based on self-reported symptoms, we consider it

descriptive and exploratory in nature, and we think that type I errors are preferable to type II errors in this setting